# The miR396–GRF Regulatory Module Controls the Embryogenic Response in *Arabidopsis* via an Auxin-Related Pathway

**DOI:** 10.3390/ijms20205221

**Published:** 2019-10-21

**Authors:** Aleksandra Szczygieł-Sommer, Małgorzata D. Gaj

**Affiliations:** Department of Genetics, University of Silesia, Faculty of Biology and Environmental Protection, 40-032 Katowice, Poland

**Keywords:** auxin, growth-regulating factor, miR396, plethora1, PLT2, somatic embryogenesis

## Abstract

In plants, microRNAs have been indicated to control various developmental processes, including somatic embryogenesis (SE), which is triggered in the in vitro cultured somatic cells of plants. Although a transcriptomic analysis has indicated that numerous *MIRNA*s are differentially expressed in the SE of different plants, the role of specific miRNAs in the embryogenic reprogramming of the somatic cell transcriptome is still poorly understood. In this study, we focused on performing a functional analysis of miR396 in SE given that the transcripts of *MIR396* genes and the mature molecules of miR396 were found to be increased during an SE culture of *Arabidopsis*. In terms of miR396 in embryogenic induction, we observed the SE-associated expression pattern of *MIR396b* in explants of the β-glucuronidase (GUS) reporter line. In order to gain insight into the miR396-controlled mechanism that is involved in SE induction, the embryogenic response of *mir396* mutants and the 35S:*MIR396b* overexpressor line to media with different 2,4-Dichlorophenoxyacetic acid (2,4-D) concentrations was evaluated. The results suggested that miR396 might contribute to SE induction by controlling the sensitivity of tissues to auxin treatment. Within the targets of miR396 that are associated with SE induction, we identified genes encoding the *GROWTH-REGULATING FACTOR* (*GRF*) transcription factors, including *GRF1, GRF4, GRF7, GRF8*, and *GRF9*. Moreover, the study suggested a regulatory relationship between miR396, *GRF*, and the *PLETHORA* (*PLT1* and *PLT2*) genes during SE induction. A complex regulatory relationship within the miR396–GRF1/4/8/9–PLT1/2 module that involves the negative and positive control of *GRF*s and *PLT* (respectively) by miR396 might be assumed.

## 1. Introduction

Somatic embryogenesis (SE), a unique plant-specific developmental process, results in the formation of somatic embryos from in vitro cultured somatic cells/tissue. Since its discovery, SE has become widely used for plant regeneration in plant biotechnology and as a model to study the totipotency of plant somatic cells in basic research. The knowledge that is gained from studies on SE and, in particular, from identifying the genetic determinants of embryogenic transition in in vitro cultured plant explants greatly contributes to improving the micropropagation and genetic transformation of plants (reviewed in Reference [1]).

Transcription factors (TFs), which play a central role in regulating gene expression, are believed to have a decisive function in the genetic reprogramming of somatic cells in SE. Accordingly, an extensive modulation of hundreds of TF genes was found during SE induction in *Arabidopsis* [2,3] and the function of several TFs in SE induction was demonstrated including, BABY BOOM (BBM [4]), LEAFY COTYLEDON2 (LEC2 [5]), WUSCHEL (WUS [6]), AGAMOUS-LIKE15 (AGL15 [7]), MYB118 [8], and EMBRYO-MAKER (EMK [9]).

In concert with TFs, microRNAs, which seem to preferentially target TF genes to control plant development [10,11], have been suggested as regulating SE induction. In line with this assumption, the differential expression of numerous miRNAs has been indicated in an embryogenic culture of *Arabidopsis*, and the function of the candidate microRNAs was attributed mainly to hormone and stress responses [12]. In addition to *Arabidopsis*, the differential expression of numerous miRNAs has been reported in the SE of other plants, including *Oryza sativa* [13], *Liriodendron tulipifera* L. *chinense* [14], *Larix laptolerix* [15], *Gossypium hirsutum* [16], *Zea mays* [17], and *Manihot esculenta* [18]. Several miRNAs have been functionally analyzed during SE, including miR160 and miR166/165 [19], miR167 [20], miR393 [21], and miR156 [22]. In addition, miR396 was postulated to regulate SE induction given the distinctly increased expression of primary (pri-miRNA) and mature-miR396 in an embryogenic culture of *Arabidopsis* [12]. In vivo, miR396 molecules have been postulated to control leaf development in *Arabidopsis* by regulating various cellular processes including the proliferation, expansion, and differentiation of cells [23,24]. In root development, the function of miR396 has been associated with regulating the switch between stem cells and transit-amplifying cells [25].

In *Arabidopsis*, miR396, which is produced from two *MIR396* (*MIR396a* and *MIR396b*) genes, has been indicated to repress the TF genes that encode the members of the *GROWTH-REGULATING FACTOR* (*GRF*) gene family [26]. In *Arabidopsis*, the *GRF* gene family comprises nine members, and miR396 recognizes the target site in seven of these genes, including *GRF1–4* and *GRF7–9* [26]. The miR396-targeted *GRF*s have been indicated to regulate plant growth and development in vivo, which involves leaf and cotyledon growth [27] and stem, flower, and root development (reviewed in Reference [28]). In the development of the *Arabidopsis* root, the GRFs act as repressors of *PLETHORA1* (*PLT1*) and *PLT2*. In turn, the PLTs activate miR396, and as a result, miR396 represses the *GRFs* [25]. Given that the increased expression of *PLT2* induces SE from *Arabidopsis* seedlings, the involvement of the miR396–GRF–PLT regulatory network in SE induction might be assumed [29].

Importantly for the possible role of miR396 in the stress-related mechanism of SE induction [29], various stress treatments (e.g., drought, salinity, low temperature, and Ultraviolet B (UVB) radiation) have been found to modulate the level of miR396 [30,31].

In order to gain insight into the function of miR396 in SE induction, we studied the embryogenic response that is induced in in vitro cultured *Arabidopsis* explants in relation to the miR396 expression level and auxin treatment. Some of the evidence implied that miR396 controls efficient SE induction through an auxin biosynthesis-related pathway, and among the targets of miR396, *GRF*s were suggested, including *GRF1/4/7/8/9*. In addition, the regulatory relationship between the miR396–GRF module and the *PLETHORA* genes, including *PLT2*, was investigated.

## 2. Results

### 2.1. miR396 Was Expressed Specifically in the SE-Induced Explant Parts

Our previous analysis of the SE transcriptome indicated a significant upregulation of *MIR396* genes and their mature product in an embryogenic culture of *Arabidopsis*, and thus, a role for miR396 in regulating SE was suggested [12]. Consistent with this assumption, the present analysis of the *MIR396b*:GUS reporter line with promoter:GUS fusions revealed that the explants that were induced toward SE preferentially accumulated the *MIR396b* transcript in the explant parts that were involved in embryogenic induction, including the cotyledons and the proximity of the shoot apical meristem (SAM) (Figure 1a–c). Moreover, we observed that, similarly to the immature zygotic embryos ((IZEs), 0 d) that were used as the explants for SE induction, somatic embryos that were developed on the SE-induced explants also showed GUS expression in the cotyledons (Figure 1a vs. Figure 1d). This result implied a similarity between the *MIR396b* expression pattern in the somatic embryos and their zygotic counterparts.

### 2.2. miR396 Contributed to SE Induction via an Auxin Biosynthesis-Related Pathway

To explore the mechanism that is involved in miR396-mediated SE induction, we analyzed the relationship between the expression level of miR396 and the embryogenic potential of the explants. To this end, explants of miR396 insertional mutants (*mir396a* and *mir396b*) and the 35S:*MIR396b* line with down- and upregulated miR396 expressions, respectively, were analyzed on induction media that had been supplemented with different concentrations (0.1, 0.5, 1.0, 10.0, and 30.0 µM) of 2,4-Dichlorophenoxyacetic acid (2,4-D), which is a synthetic auxin that is commonly used to induce SE. The results showed that on a standard medium with 5 µM 2,4-D, which is highly efficient for SE induction in a wild type (WT; Col-0)) culture, the mutants and the overexpression of the 35S:*MIR396b* line displayed a decreased and increased SE response, respectively, compared to the WT explants (Figure 2). However, on media that had been supplemented with 2,4-D at concentrations that were suboptimal for SE induction in a WT culture (0.1, 0.5, and 1.0 µM), the embryogenic response of the *mir396* mutants was distinctly higher than in the WT explants (Figure 2a–d). In contrast, the explants of the 35S:*MIR396b* overexpressor line were opposite to the mutants’ phenotypes. Accordingly, the explants overexpressing miR396 showed a distinctly higher embryogenic response after treatment with a 2,4-D concentration that was overoptimal for SE induction in WT culture (Figure 2e,f). Altogether, the results showed a negative relationship between the miR396 expression level and the sensitivity of the explants to exogenous auxin treatment, and thus, we hypothesize that miR396 may contribute to SE induction via an auxin-related pathway.

In order to verify this assumption, we evaluated the content of the indolic compounds in relation to the miR396 expression level. An analysis of the *mir396a* and *mir396b* mutants showed as much as a 1.85- and 1.50-fold increased level of indolic compounds, respectively (Figure 3a; Appendix A). In contrast to the mutant cultures, the level of indolic compounds was decreased by as much as two-fold in the 35S:*MIR396b* culture. Given that Indolilic-3-acetic acid (IAA) contributed to the pool of indolic compounds, these results suggest a negative impact of the miR396 expression on the IAA accumulation in the SE-induced explants, and thus we assume that miR396 might control the auxin biosynthesis pathway during SE induction.

Thus, we evaluated the expression level of the *YUC1, YUC4*, and *YUC10* genes encoding the key enzymes of the auxin biosynthesis pathway during SE in *Arabidopsis* [32]. Analyses of the *mir396* mutants and the 35S:*MIR396b* line suggested a relationship between *YUC1* and *YUC4* and miR396 during SE induction, and accordingly, the gene transcription was substantially up- and downregulated in the cultures of the *mir396* mutants and the 35S:*MIR396b* line, respectively (Figure 3b,c). In contrast to *YUC1* and *YUC4*, the expression of *YUC10* was not detected in the cultures with a deregulated miR396 expression.

In conclusion, the results imply that miR396 might contribute to the SE response via negative control of the *YUC1* and *YUC4* genes of the auxin biosynthesis pathway and to a decrease in the endogenous auxin accumulation.

### 2.3. miR396 Controlled the GRF Genes during SE Induction

To gain more insight into the function of miR396 in SE induction, we profiled the expression of the *GRF* genes, which are potential targets of miR396 in plants [29]. The results of an RT-qPCR analysis showed a significant downregulation of all of the analyzed *GRFs* (*GRF1–9*) during the early (5 d) and advanced (10 d) stages of the SE culture that had been induced in the WT (Col-0) explants (Figure 4a). A significantly decreased level of the *GRF1–9* transcripts in SE was found to be opposite to the distinct accumulation of miR396 during SE (Appendix A), and this observation suggests a regulatory relationship between miR396 and the *GRF* genes.

In order to verify this assumption, we analyzed the expression of the *GRF* genes in the SE-induced explants of the *mir396* (*mir396a* and *mir396b*) mutants and the 35S:*MIR396b* overexpressor line. The analysis showed that the transcripts of five of the analyzed genes (*GRF1, GRF4, GRF7, GRF8* and *GRF9*) were significantly up- and down-regulated in the *mir396* mutants and *MIR396b* overexpressor line culture, respectively (Figure 4b–f). In contrast, there was no impact of the miR396 expression on the transcript level of *GRF2, GRF3, GRF5*, and *GRF6* (Appendix A). In conclusion, the reverse expression level of *GRF1, GRF4, GRF7, GRF8*, and *GRF9* in the *mir396* mutants versus the 35S:*MIR396b* overexpressor line cultures implies that miR396 might control these *GRFs* during SE induction.

To confirm that the miR396-regulated *GRFs* contributed to SE induction via an auxin-related mechanism, we evaluated the embryogenic response of the 35S:*GRF1* transgenic line and the *grf4* mutant in media with different concentrations of 2,4-D. The analyses indicated that the 35S:*GRF1* explants that had been treated with an auxin concentration of 1.0 µM, which is suboptimal for SE induction in a WT culture, displayed the highest level of SE efficiency and productivity (Figure 5a,b). This finding implies that similarly to the *mir396* mutants with a significantly impaired miR396 accumulation (Figure 2a–d), the overexpression of *GRF1* resulted in an increased sensitivity of tissues to the auxin treatment that was effective for SE induction.

Consistent with this result, the *grf4* mutant phenotype was found to be opposite to the 35S:*GRF1*, and the highest embryogenic response of the mutant explants was observed on the medium with 10.0 µM of 2,4-D, i.e., a concentration that was overoptimal for SE induction in a WT culture (Figure 5c,d). Moreover, we observed that the response of the *grf4* mutant to auxin treatment was similar to the one that was observed in the culture overexpressing *MIR396b*, in which the overoptimal 2,4-D concentration (10.0 µM) increased the SE response (Figure 2e,f). In contrast to the SE-affected phenotype of *grf4,* the embryogenic potential of the *grf1, grf8*, and *grf9* knockdown mutants was not significantly impaired (Appendix A), and that could suggest a functional redundancy between these *GRFs* in SE.

Taken together, the results suggest that through the repression of *GRF1* and *GRF4*, miR396 might control the embryogenic transition by modulating the sensitivity of tissues to auxin treatment. We also assumed that *GRF7*, *GRF8*, and *GRF9* might be the targets of miR396 in SE, and further analyses using the relevant mutant lines are required to verify this regulatory relationship.

### 2.4. The Regulatory Relationship between miR396 and PLT1/PLT2 in SE Induction

A regulatory interaction between miR396, *GRF*, and *PLTETHORA* (*PLT*) transcription factors in root development [25] motivated us to investigate the relationship between *PLT* genes and the miR396/GRF module during SE. We found that *PLT2* positively regulated SE induction because the cultures of the *plt2* mutant and the 35S:*PLT2*-GR overexpressor line displayed a significantly reduced and enhanced, respectively, SE response (Figure 6a–c).

To further explore the relationship between PLT and miR396 in SE induction, we profiled the expression of *PLT1* and *PLT2* in the Col-0 (WT) explants that had been cultured on a standard auxin (E5) medium. We observed that the *PLT1* and *PLT2* expression patterns reflected those of miR396 (Appendix A). In addition, we found the *PLT1* and *PLT2* transcript levels to be down- and upregulated in the cultures of the *mir396* mutants and the 35S:*MIR396b* overexpressing line, respectively (Figure 7a,b). These results suggested a positive relationship between the level of miR396 and the PLT1/PLT2 expression during SE induction. In addition, the *plt2* mutant culture displayed a significantly decreased miR396 level (Figure 7c) and the upregulation of *GRF1, GRF4, GRF8*, and *GRF9* transcripts (Figure 7d).

Altogether, the results suggest a role for the miR396–GRF1/4/8/9–PLT1/2 module in SE induction in *Arabidopsis*. Complex regulatory relationships within this module that involve negative and positive control over GRFs and PLT (respectively) by miR396 might be assumed.

## 3. Discussion

### 3.1. An Accumulation of miR396 Was Associated with SE Induction

Stress responses play a central role in the molecular mechanism of SE induction [29,33]. Given that stresses have been reported to distinctly upregulate miR396 [30,31], the increased accumulation of miR396 in the embryogenic culture of *Arabidopsis* [12] might reflect the explant’s response to stress that is imposed in vitro by different culture conditions, including treatment with 2,4-D [29]. Accordingly, we found that in response to 2,4-D treatment, the miR396 transcripts were specifically expressed in the SE-involved areas of the explants, i.e., in proximity to cotyledons and the SAM [34]. Thus, to gain insight into the biological function of miR396 in SE induction, we explored the relationship between miR396 and 2,4-D, which is a synthetic auxin that has an SE-promoting activity in *Arabidopsis* and other plants [35].

### 3.2. miR396 Controlled the Embryogenic Response by Modulating the Sensitivity of Tissues to Auxin

Numerous hormone-related miRNAs have been suggested to control SE induction in *Arabidopsis* [12], and among them, miR393, miR160, and miR165/166 have been indicated to impact auxin metabolism and signaling in embryogenic explants [19,21]. A relationship between miR396 and hormones during SE might also be expected given that ethylene has been found to modulate the expression of miR396 in *Medicago truncatula* roots [36] and that the involvement of miR396 in brassinosteroid (BR) and gibberellin (GA) signaling in rice plants [37] has also been reported.

Here, we provide evidence of the relationship between miR396 and the auxin that accumulated in the SE-induced explants of *Arabidopsis* [32]. Accordingly, an inverse relationship between the miR396 expression level and the sensitivity of the explants to auxin treatment was demonstrated (Figure 2), and a negative impact of the miR396 expression on the accumulation of the indolic compounds (including IAA in the SE-induced explants) was indicated (Figure 3a). Consistently, two *YUC* genes (*YUC1* and *YUC4*) of the auxin biosynthesis pathway that contribute to the embryogenic response in *Arabidopsis* [32] were found to be miR396-repressed during SE induction (Figure 3b,c). In agreement with the assumption that miR396 negatively controls auxin biosynthesis during SE induction in *Arabidopsis* explants, miR396 was found to repress auxin biosynthesis during the development of *Oryza sativa* plants [38]. We observed that the level of indolic compounds in the 35S:*MIR396b* culture was not reduced on day 10 despite a decreased expression level of *YUC* genes. This may suggest that besides a tryptophan-dependent and YUC-involved auxin biosynthesis pathway, a tryptophan-independent pathway might also contribute to IAA production during an advanced SE stage (10 d), which is related to somatic embryo development. In support of this assumption, an important role of the tryptophan-independent auxin biosynthesis pathway in the establishment of the apical–basal pattern during early zygotic embryogenesis has been indicated in *Arabidopsis* [39].

### 3.3. miR396 Regulated SE Induction by Repressing GRFs (GRF1, 4, 7, 8, and 9)

Members of the *GRF* gene family have been postulated as candidate targets of miR396, and a role for the miR396–GRF module in *Arabidopsis* development has been reported, including the regulation of cell proliferation in leaves and the transition of stem cells in roots [23,25]. In addition, miR396 has been indicated to control *GRFs* in other plants, including seed development in rice and barley [39] and the formation of adventitious roots in apple rootstock [40].

Consistent with these findings, our results imply that miR396 targeted the *GRFs* in order to regulate the embryogenic transition that was induced in in vitro cultured explants of *Arabidopsis*. Conclusive for this postulate was the negative relationship between the accumulation of miR396 and the expression level of five of the *GRF* genes (*GRF1*, *4*, *7*, *8*, and *9*) in the embryogenic culture of the *mir396* mutants and the 35S:*MIR396b* line (Figure 4b–f). In support of the postulated engagement of the miR396–GRF module in the embryogenic transition in *Arabidopsis*, the miR396-controlled repression of *GRF1, GRF4*, and *GRF8* has been reported during SE of *Lilium pumilum* [41], *hybrid yellow poplar* [14], and cotton [16]. A report on the *GRF4*-regulated development of cotyledons and SAM in zygotic embryos that provide SE-responsive tissue in *Arabidopsis* [34,42] also supported a role for *GRFs* in embryogenic development. Relevant to the stress and cell reprogramming-associated mechanism of SE induction [33], the miR396–GRF1 module was found to control the specification and differentiation of cells during the stress response in *Arabidopsis* plants [43].

This study provides evidence that the miR396–GRF module contributes to SE induction via an auxin-related mechanism. In line with this assumption, an analysis of the 35S:*GRF1* explants in media with different concentrations of 2,4-D showed that the embryogenic response of these explants phenocopied that of the *mir396* mutants but contrasted with the response of the *MIR396b* overexpression line (Figure 5a,b). Moreover, the explants of the *grf4* mutant had a decreased sensitivity to auxin treatment that was also characteristic of the 35S:*MIR396b* overexpression culture (Figure 5c,d). A lack of the SE-affected phenotype in the other analyzed *grf* mutants (Appendix A) might suggest a functional redundancy of some *GRFs* in the control of SE, as was indicated for *GRF1–3* in the development of leaves and cotyledons in vivo [27]. In contrast, the distinct SE phenotype of the *grf4* mutant (Figure 5b) seemed to confirm an important function of *GRF4* in plant embryogenesis given its reported role in the development of cotyledons and the SAM in the zygotic embryogenesis of *Arabidopsis* [42]. In support of the contribution of *GRF4* to SE, the cotyledons and SAM of a zygotic embryo provide highly responsive tissue in SE induction in *Arabidopsis* [34].

Collectively, these results imply that through the repression of *GRFs*, including *GRF1*, *4*, *7*, *8*, and *9*, miR396 appeared to regulate the auxin-related mechanism of the embryogenic transition that was induced in *Arabidopsis* explants.

### 3.4. miR396 Positively Controlled PLT Genes in SE Induction via GRFs

In order to dissect the biological function of the miR396–GRF module in the SE-associated regulatory network of genes, we found it reasonable to gain insight into the relationship of this module with the TFs that have a documented regulatory role in the embryogenic transition. Among the TFs that control embryogenic induction in somatic plant cells, the *BABY BOOM* (*BBM*) gene, which is a member of the *PLETHORA* (*PLT*) clade of genes that encode AP2/ERF TFs, has been identified [4]. We postulated that miR396 might indirectly control *BBM/PLT* genes through *GRFs* during SE induction. The lack of miR396-targeted sequences in the *PLT* genes [26] rules out a direct miR396–PLT regulatory relationship. A role for the miR396–PLT–GRF node in SE induction was postulated given that the expression pattern of *PLT1*/*PLT2* in the embryogenic culture was found to be similar to miR396 (Appendix A) and that the decreased expression of *PLT1/PLT2* in the *mir396* mutants contrasted with the accumulation of the *PLT1/PLT2* transcripts in the 35S:*MIR396b* culture (Figure 7a,b). In addition, a decreased expression level of the *PLT* genes in the *GRF2* and *GRF3* overexpression lines was reported during root development [25]. *GRF1* was reported to have a redundant function to *GRF2* and *GRF3* [27], and thus a similar regulatory relationship between *GRF1* and *PLT* might be assumed. Further evidence for a positive regulatory impact of miR396 on *PLTs* included the similar sensitivity to auxin treatment of the miR396 and *plt2* mutant explants, which showed enhanced SE induction after treatment with a 2,4-D concentration that was suboptimal for the WT culture (Appendix A).

In order to control the transition of root stem cells into amplifying cells in the root meristem of *Arabidopsis*, miR396 negatively regulates *GRFs*, which are repressors of *PLTs* (*PLT1/PLT2/BBM*), which in turn are necessary to activate the *MIR396* genes [25]. We postulate that a congruous regulatory feedback loop between miR396, *GRFs*, and *PLTs* might operate during SE because the embryogenic culture of the *plt2* mutant showed a decreased level of miR396 and an increased expression of *GRFs* (Figure 7c,d).

Taken together, we postulate that through the repression of *GRFs* (*GRF1, GRF4, GRF8*, and *GRF9*), miR396 activates *PLTs* (*PLT1/PLT2/BBM*). Given that the *PLT* genes (*BBM* and *PLT2*) have been demonstrated to trigger SE through the direct transcriptional regulation of the LEC1–ABI3–FUS3–LEC2 network that controls embryo identity [44], we also assumed that *PLTs* would impact auxin accumulation in the SE-induced explants by controlling the *LEC2* gene. In support of the positive control of *LEC2* by *PLT*, the *plt2* mutant showed a significantly decreased level of *LEC2* transcripts (Appendix A).

In addition to *PLTs*, our analysis of the *MIR396* promoter sequence [45] revealed that other key regulators of SE induction, including *AGL15* [7], might directly regulate miR396 embryogenic induction. *AGL15* controls multiple hormone interactions during SE induction, including the direct targeting of *LEC2*, which is an activator of auxin biosynthesis in SE [32,46]. In support of a relationship between *AGL15* and the miR396–GRF module, *GRF9* was identified as being a possible direct target of *AGL15*, and *GRF1* was postulated to control *PHAVOLUTA* (*PHV*) [47], which positively controls the LEC2-mediated pathway of SE induction in *Arabidopsis* [19,32]. In summary, complex and versatile regulatory interactions seem to operate between miR396 and the TFs that play a central role in SE induction (*BBM, AGL15*, and *LEC2*). In the postulated model, *LEC2* seemed to be a key link between the GRF/PLT and auxin-regulated SE induction, and further experiments are required to verify an miR396-controlled network that is involved in auxin-regulated embryogenic transition in *Arabidopsis* (Figure 8).

## 4. Materials and Methods

### 4.1. Plant Material and Growth Conditions

The seeds of different genotypes of *Arabidopsis thaliana* (L.) Heynh., including the Columbia (Col-0) WT genotype and the insertional mutants *mir396a* (N447416), *mir396b* (N412157), *grf4* (N657589), and *plt2* (N676869), were supplied by NASC (The Nottingham Arabidopsis Stock Centre, Nottingham, UK). The seeds for the 35S:*MIR396b* and MIR396b/GUS reporter lines were kindly provided by Javier F. Palatnik (Research Council, Institute of Molecular and Cell Biology in Rosario, Argentina). The seeds for the 35S:*GRF1* transgenic line, which was derived from the Wassilewskija (WS) genotype, were kindly provided by Jeong Hoe Kim (Department of Biology, Kyungpook National University, Daegu, South Korea). The transgenic line (overexpressing) *PLT2* gene (35S:*PLT2*-GR), was kindly provided by Kim Boutilier (Wageningen Plant Research, Wageningen, Netherlands). Descriptions of the transgenic genotypes that were used in the study are presented in Appendix A. The level of mature miR396 was significantly decreased and increased in the analyzed *miR396* mutants and 35S:*MIR396b* line, respectively (Appendix A).

The plants were kept in a growth chamber at 21 ± 1 °C under a 16/8-h photoperiod of 40 μM m^−2^s^−1^ white fluorescent light.

### 4.2. Somatic Embryogenesis Induction

Immature zygotic embryos (IZEs) from different genotypes at the green cotyledonary stage were used as the explants for the in vitro cultures. To induce SE, the standard protocol was used [48]. The IZEs were cultured on an E5 solid medium containing B salts and vitamins [49] that was supplemented with 5 µM 2,4-dichlorophenoxyacetic acid (2,4-D, Sigma-Aldrich, St. Louis, MO, USA), 20 g L^−1^ sucrose, and 8 g L^−1^ agar. In some of the experiments, different concentrations of 2,4-D (0.1, 0.5, 1.0, 10, and 30 µM) were used in the induction medium.

At selected time points of the culture (0, 5, and 10 days), the explants of Col-0, *mir396a*, *mir396b*, 35S:*MIR396b*, and *plt2* were sampled for transcriptome analysis.

The explant capacity for SE was evaluated on the 21st day of the in vitro culture. Two parameters were calculated: SE efficiency, i.e., the percentage of the explants producing somatic embryos, and SE productivity, i.e., the average number of somatic embryos produced per explant. All of the culture combinations were estimated in three replicates: at least 30 explants (10 explants per Petri dish) were analyzed per 1 replicate.

### 4.3. Content of Indolic Compounds

To evaluate the content of the indolic compounds, including IAA, the colorimetric technique was used [50]. Explants of the Col-0, *mir396a*, and *mir396b* mutants and 35S:*MIR396b* were induced for 5 and 10 days on an E5 medium. The concentration of IAA was established using a calibration curve of pure IAA as the standard following a linear regression analysis. Each measurement was performed in three replicates.

### 4.4. RNA Isolation and RT-qPCR Analysis

An Ambion RNAqueous Kit (Thermo Fisher Scientific, Waltham, MA, USA) was used to isolate the total RNA from the IZE explants on days 0, 5, and 10 of the culture on an auxin (E5) medium in three biological repeats. The concentration and purity of the RNA were evaluated using an ND-1000 spectrophotometer (Nano Drop Technologies, LLC, Wilmington, DE, USA). Total RNA that had been treated with RQ1 RNase-free DNase I (Promega Corporation, Madison, WI, USA) was reverse-transcribed using a RevertAid First-Strand cDNA Synthesis Kit (Thermo Fisher Scientific, Waltham, MA, USA) following the manufacturer’s instructions. A LightCycler^®^ 480 SYBR Green I Master (Roche, Basel, Switzerland) was applied for the RT-qPCR reactions. The primers that were used for the expression profiling of the studied genes are listed in Appendix A.

### 4.5. Stem-Loop RT-PCR for Mature miRNA Detection

A mirVana™ miRNA Isolation Kit was used to isolate the small RNAs from the IZE explants that were induced on an auxin (E5) medium on days 0, 5, and 10 of the culture. The concentration and purity of the small RNAs were evaluated using an ND-1000 spectrophotometer (Nano Drop Technologies, LLC, Wilmington, DE, USA). The design of the oligonucleotides, stem-loop reverse transcription, and real-time qPCR were performed according to Speth and Laubinger [51]. The real-time qPCR analysis for the mature miR396 accumulation was performed using a LightCycler 480 (Roche, Basel, Switzerland). The primers that were used to detect miR396 are listed in Appendix A.

### 4.6. Transcript Level Calculation

The relative transcript levels were calculated and normalized to an internal control, the *At4g27090* gene-encoded 60S ribosomal protein. The plant tissues for real-time qPCR analysis were produced in three biological repetitions, and two technical replicates of each repetition were performed. The relative expression level was calculated using the 2^−∆∆*C*t^ method [52].

### 4.7. Histochemical Staining of GUS

The histochemical analysis of GUS activity was performed as described by Jefferson et al. (1987). The explants were stained in X-Gluc (5-bromo-4-chloro-3-indolyl β-D-glucuronide-cyclohexylammonium salt) (Sigma Aldrich, St. Louis, MO, USA). The reaction was carried out in the dark at 37 °C for 12 h. The GUS signal was visualized using a Zeiss Stemi 2000-C microscope (Zeiss, Oberkochen, Germany).

### 4.8. Statistical Analysis

Student’s *t*-test was used to calculate any significant differences (at *p* = 0.05) between the compared samples. The figures present the averages with standard deviations.

## Figures and Tables

**Figure 1 ijms-20-05221-f001:**
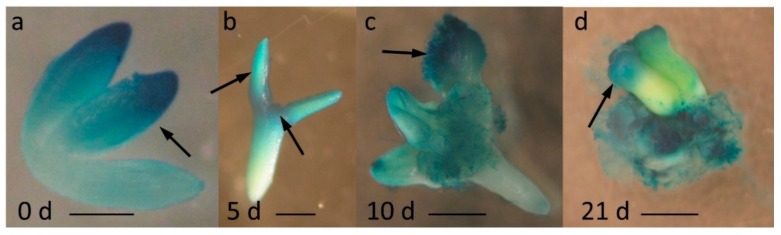
The spatiotemporal expression pattern of *MIR396b* in immature zygotic embryo (IZE) explants that were cultured for (**a**) 0, (**b**) 5, (**c**) 10, and (**d**) 21 days on a standard somatic embryogenesis (SE) induction medium (E5). Cotyledons and the shoot apical meristem (SAM) area of the IZE, i.e., the explant parts that were involved in SE induction, showed a β-glucuronidase (GUS) signal (**a**–**c**). Similarly to the IZEs (**a**), the somatic embryos (**d**) had a GUS signal in the cotyledons. Arrows point to the GUS signal. Scale bar: 0.2 mm (**a**,**b**), 1 mm (**c**,**d**).

**Figure 2 ijms-20-05221-f002:**
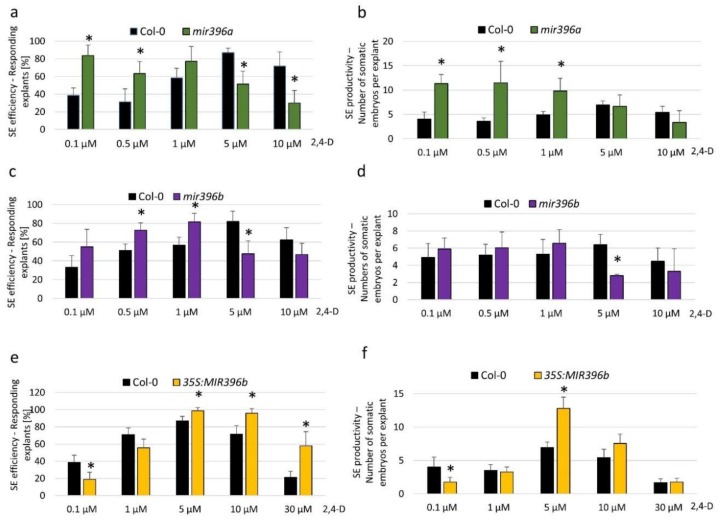
The embryogenic potential of the explants in relation to the miR396 expression level and auxin treatment. Explants of Col-0 (wild type - WT): (**a**,**b**) *mir396a*, (**c**,**d**) *mir396b*, and (**e**,**f**) 35S:*MIR396b* were cultured on an SE induction medium that had been supplemented with different concentrations of 2,4-Dichlorophenoxyacetic acid (2,4-D). The efficiency (**a**,**c**,**e**) and productivity (**b**,**d**,**f**) of the SE were evaluated. Values that were significantly different from those of the WT culture are indicated with asterisks (*) (*p* < 0.05; *n* = 3 ± SD).

**Figure 3 ijms-20-05221-f003:**
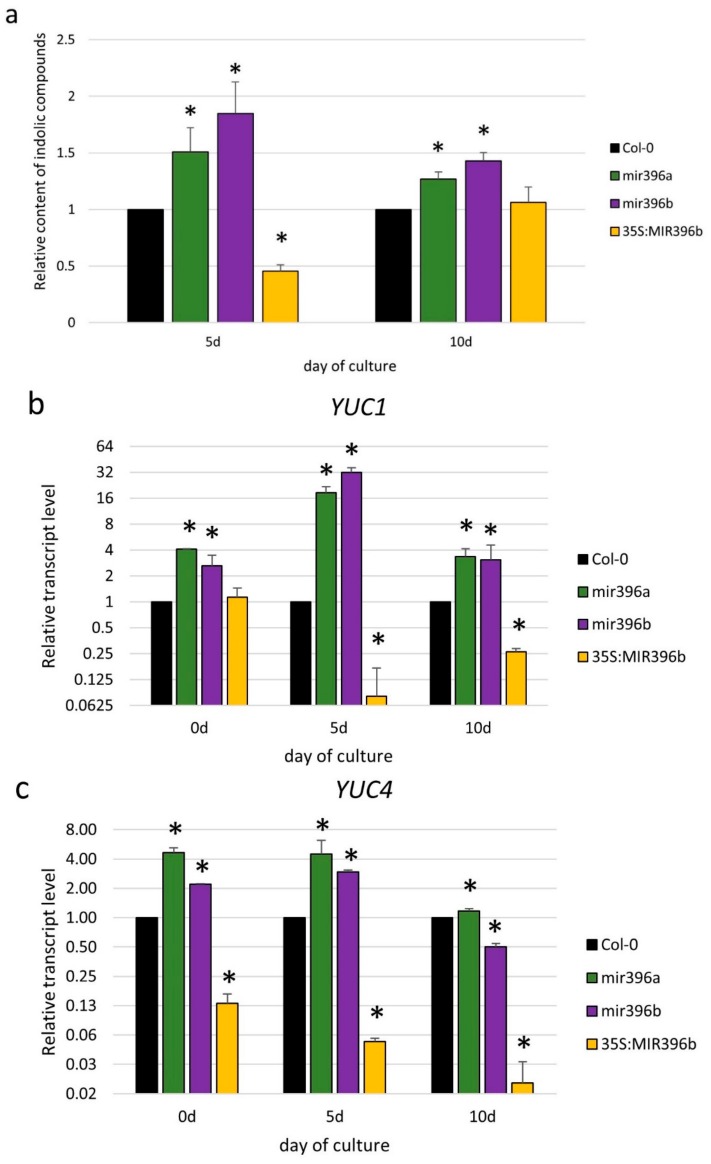
The auxin-related function of miR396 in an SE culture. Explants of *mir396a*, *mir396b*, 35S:*MIR396b*, and Col-0 (WT) were cultured on a standard SE induction medium (E5). (**a**) The relative content of the indolic compounds and the expression level of the auxin biosynthesis of the (**b**) *YUC1* and (**c**) *YUC4* genes were evaluated in the SE-induced explants. The relative transcript level was normalized to the internal control (*At4g27090*) and calibrated to a Col-0 culture of the same age. The bars represent the standard deviation. Values that were significantly different from those of a WT culture of the same age are indicated with asterisks (*) (*p* < 0.05; *n* = 3 ± SD).

**Figure 4 ijms-20-05221-f004:**
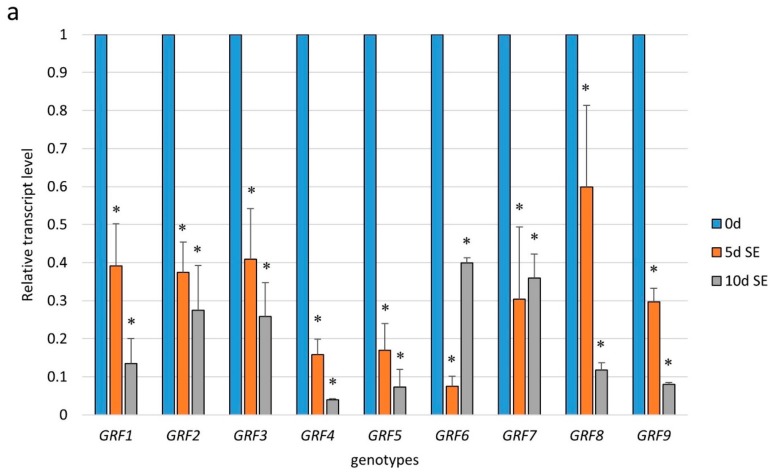
The regulatory relationship between miR396 and the *GRF* genes during SE. The expression of (**a**) the *GRF* genes in the SE culture of Col-0 and (**b**) *GRF1*, (**c**) *GRF4*, (**d**) *GRF7*, (**e**) *GRF8*, and (**f**) *GRF9* in the SE cultures of the *mir396a*, the *mir396b* mutants, and the 35S:*MIR396b* line. The explants were induced on a standard SE induction medium (E5). The relative transcript level was normalized to the internal control (*At4g27090*) and calibrated to a WT culture of the same age. Values that were significantly different from those of a WT culture of the same age are indicated with asterisks (*) (*p* < 0.05; *n* = 3 ± SD).

**Figure 5 ijms-20-05221-f005:**
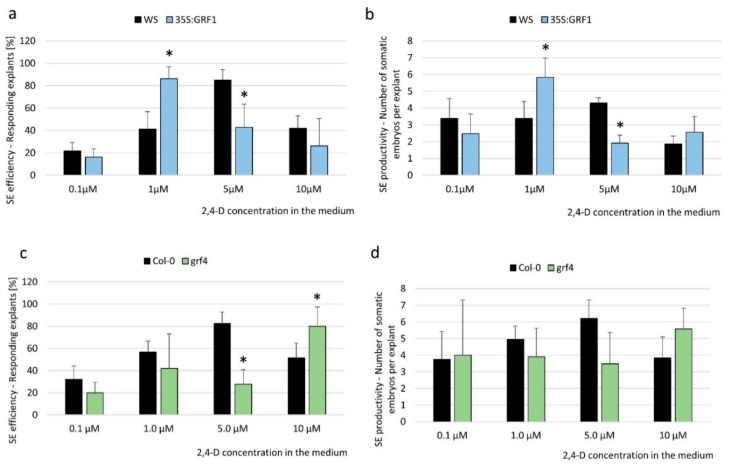
*GRF1* and *GRF4* controlled the embryogenic response of the explants that had been cultured on the auxin media. Explants of (**a**,**b**) the 35S:*GRF1* line, (**c**,**d**) the *grf4* mutant, and their WT genotypes, (Wassilewskija (WS) and Col-0, relevantly) were cultured on an SE induction medium that had been supplemented with different concentrations of 2,4-D. (**a**,**c**) The efficiency and (**b**,**d**) productivity of SE were evaluated in a 21-day-old culture. Values that were significantly different from those of the WT culture are indicated with asterisks (*) *p* < 0.05; *n* = 3 ± SD).

**Figure 6 ijms-20-05221-f006:**
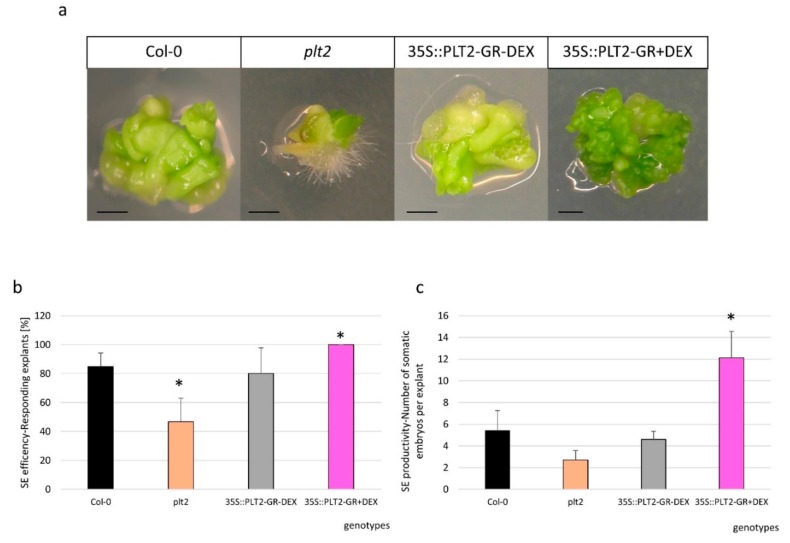
*PLT2* controlled the embryogenic response of the explants that were cultured on the auxin medium. (**a**) The explants of Col-0 (WT), the *plt2* mutant, and the 35S:*PLT2*-GR line were cultured for 21 days on a standard E5 medium, after which the levels of (**b**) efficiency and (**c**) productivity of SE were evaluated. Dexamethasone (DEX) treatment (+DEX) was used to induce *PLT2* overexpression in the 35S:*PLT2*-GR explants (35S:PLT2-GR + DEX). The bars represent the standard deviation. Values that were significantly different from those for the WT culture are indicated with asterisks (*) (*p* < 0.05; *n* = 3 ± SD). Scale bar: 1 mm.

**Figure 7 ijms-20-05221-f007:**
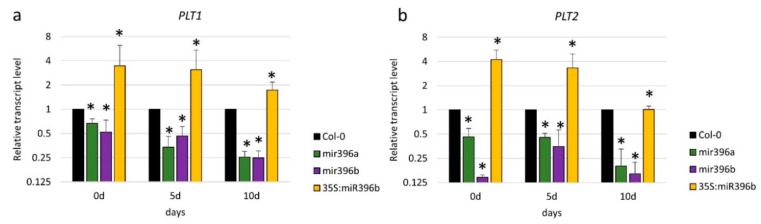
The regulatory relationship within the miR396–GROWTH REGULATING FACTOR (GRF)–PLETHORA (PLT) module. (**a**) The expression level of the *PLT1* and (**b**) *PLT2* genes in the cultures of the *mir396* mutants and the 35S:*MIR396b* line; (**c**) the decreased accumulation of miR396 in the culture of the *plt2* mutant; (**d**) the increased expression level of *GRFs* (*GRF1, GRF2, GRF3, GRF4, GRF7, GRF8*, and *GRF9*) in the culture of the *plt2* mutant. The explants were cultured on a standard E5 medium. The relative transcript level was normalized to the internal control (*At4g27090*) and calibrated to a WT (Col-0) culture of the same age. The bars represent the standard deviation. Values that were significantly different from those of the WT culture are indicated with asterisks (*) (*p* < 0.05; *n* = 3 ± SD).

**Figure 8 ijms-20-05221-f008:**
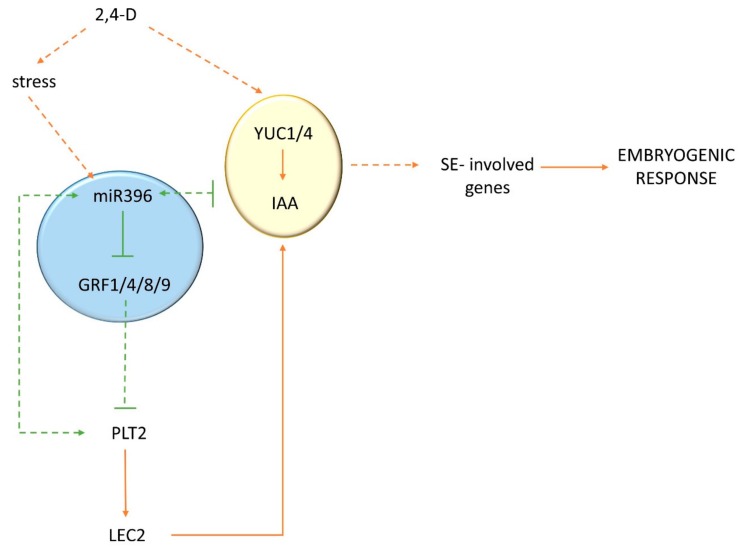
A model of the miR396-controlled network that is involved in auxin-regulated SE induction in *Arabidopsis*: 2,4-D treatment increased the accumulation of miR396 in the explants by activating the auxin-responsive *MIR396* genes. As a result, the *GRF* genes (*GRF1/4/8/9*) were repressed, the *PLT* genes (*PLT1/PLT2/BBM*) were activated, and, in turn, the *LEC2* gene, which is a positive regulator of the *YUC* (*YUC1* and *YUC4*) genes that are engaged in auxin biosynthesis during SE, was upregulated [25,32,44,47]. A feedback regulatory loop between *PLT* and miR396 might also be expected [25].

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
