# Peer review of "The miR396–GRF Regulatory Module Controls the Embryogenic Response in Arabidopsis via an Auxin-Related Pathway"

_ijms, 2019, doi:10.3390/ijms20205221_

Round 1
Reviewer 1 Report
The authors have clarified questions I raised in my previous review. It presents a nice work on miR396-GRF regulatory module during somatic embryogenesis induction in Arabidopsis.
Reviewer 2 Report
Authors responded adequately to the comments. I suggest the article to be published in IJMS
This manuscript is a resubmission of an earlier submission. The following is a list of the peer review reports and author responses from that submission.
Round 1
Reviewer 1 Report
The work of Szczygieł-Sommer and Gaj presents a significant amount of data demonstrating the role of the miR396-GRF regulatory module on embryonic response in Arabidopsis. The work is solid and the conclusions derived are appropriate.
Just few minor points:
Abstract. p.1/lines 14-16: Please revise for clarity. p.2/line 46: “are believed to regulate SE induction”. Please change “believe” with “suggested” or “demonstrated”. p.2/line 76-77: Please revise for clarity. p.2/line 83: It is not “signal”. I suggest “expression”. Figure 1. Please incorporate scale bars.Reviewer 2 Report
The role of miR396 in SE induction was studied in this manuscript. Regulatory network of miR396-GRFs-PLT module and their function in SE response were investigated using overexpressed lines and mutants.
- The introduction needs to be restructured by enriching background of miR396-GRFs-PLT regulatory network.
- As there are two MIR396 genes in Arabidopsis, the reason of choosing MIR396b for spatio-temporal expression and overexpression was not mentioned, even though mir396a mutant displayed an increased SE productivity but not the mir396b mutant (Figure 2 b,d).
- Using colorimetric technique, the content of indolic compounds was measured in this study. Even IAA contributes to the pool of indolic compounds, these is not enough evidence to show that IAA accumulation was affected, doesn’t need to mention the effect of miR396 expression here.
- miR396 negatively control YUC1 and YUC4 genes of auxin biosynthesis pathway was suggested (Figure 3), but the repression of YUC1 and YUC4 in 35S:MIR396b line didn’t lead to reduction of indolic compounds on the 10th day of SE induction. This point should be discussed.
- To confirm the contribution of miR396-regulated five GRFs in SE induction, 35S:GRF1 transgenic line and the grf4 mutant were inspected. It is unaccountable that only one of five GRFs was chosen for overexpression or knockout even not the same one.
- To clear the regulatory relationship among miR396, GRFs and PLT in SE induction, more evidence is needed. What is the expression of PLT genes in the 35S:GRF1 transgenic line and the grf4 mutant? Does miR396 regulate PLT directly or indirectly by GRFs or both? Moreover as the model of miR396-controlled network shown on Figure 8, LEC2 is the key link between GRF/PLT and auxin-regulated SE induction.
Reviewer 3 Report
The paper is an excellent study on the function of miR396 in in vitro induction of somatic embryos in Arabidopsis. The study is well planned and demonstrates that the miR396 contributes to somatic embryogenesis induction via complex auxin-related mechanism, including modulation of the cells sensitivity to auxins, repression of by repressing growth regulating factors (GRF1, 4, 7, 8 and 9) and indirect positive induction of PLETHORA genes (PLT1 and PLT2). The experiments are well organized and executed with the necessary replicates. The data is presented on clear and logical way and the fully support the conclusions made by the authors. The results are reliable, and confirmed by all necessary statistical analyses. In my opinion, the paper is important contribution in understanding the functions of miRNAs in plants development, and I think it would be of interest to the readers.
There are some minor suggestions I would like to make to the authors:
L81-82 – replace “immature zygotic embryos” with “immature zygotic embryos (IZE)”;
L86 – add a scale bar in photos of Figure 1;
L125 – figure 3b –double check the significant differences of mir396b at day 10 – according to me it seems to be significantly different from the Col-0.
L139-140 – show the chart with the expression level of YUC10 as well;
L200 – Figure 6a – add scale bars;